# Surface Modification of Fumed Silica by Plasma Polymerization of Acetylene for PP/POE Blends Dielectric Nanocomposites

**DOI:** 10.3390/polym11121957

**Published:** 2019-11-28

**Authors:** Xiaozhen He, Ilkka Rytöluoto, Rafal Anyszka, Amirhossein Mahtabani, Eetta Saarimäki, Kari Lahti, Mika Paajanen, Wilma Dierkes, Anke Blume

**Affiliations:** 1Elastomer Technology and Engineering, Department of Mechanics of Solids, Surfaces & Systems (MS3), Faculty of Engineering Technology, University of Twente, P.O.box 217, 7500 AE, Enschede, The Netherlands; a.mahtabani@utwente.nl (A.M.); a.blume@utwente.nl (A.B.); 2VTT Technical Research Centre of Finland Ltd., FI-33101 Tampere, Finland; Ilkka.Rytoluoto@vtt.fi (I.R.); Eetta.Saarimaki@vtt.fi (E.S.); Mika.Paajanen@vtt.fi (M.P.); 3High Voltage Engineering, Tampere University, FI-33720 Tampere, Finland; kari.lahti@tuni.fi

**Keywords:** acetylene plasma modification, nano silica, dispersion, PP, POE, dielectric nanocomposites, TSDC

## Abstract

Novel nanocomposites for dielectric applications-based polypropylene/poly(ethylene-co-octene) (PP/POE) blends filled with nano silica are developed in the framework of the European ‘GRIDABLE’ project. A tailor-made low-pressure-plasma reactor was applied in this study for an organic surface modification of silica. Acetylene gas was used as the monomer for plasma polymerization in order to deposit a hydrocarbon layer onto the silica surface. The aim of this modification is to increase the compatibility between silica and the PP/POE blends matrix in order to improve the dispersion of the filler in the polymer matrix and to suppress the space charge accumulation by altering the charge trapping properties of these silica/PP/POE blends composites. The conditions for the deposition of the acetylene plasma-polymer onto the silica surface were optimized by analyzing the modification in terms of weight loss by thermogravimetry (TGA). X-ray photoelectron spectroscopy (XPS) and energy-dispersive X-ray fluorescence spectroscopy (EDX) measurements confirmed the presence of hydrocarbon compounds on the silica surface after plasma modification. The acetylene plasma modified silica with the highest deposition level was selected to be incorporated into the PP/POE blends matrix. X-ray diffraction (XRD) showed that there is no new crystal phase formation in the PP/POE blends nanocomposites after addition of the acetylene plasma modified silica. Differential scanning calorimetry results (DSC) show two melting peaks and two crystallization peaks of the PP/POE blends nanocomposites corresponding to the PP and POE domains. The improved dispersion of the silica after acetylene plasma modification in the PP/POE blends matrix was shown by means of SEM–EDX mapping. Thermally stimulated depolarization current (TSDC) measurements confirm that addition of the acetylene plasma modified silica affects the charge trapping density and decreases the amount of injected charges into PP/POE blends nanocomposites. This work shows that acetylene plasma modification of the silica surface is a promising route to tune charge trapping properties of PP/POE blend-based nanocomposites.

## 1. Introduction

Space charge accumulation is a major problem for insulation materials, particularly for high voltage direct current (HVDC) applications. Space charge can accumulate in specific regions in the bulk of the material resulting from the imbalance between the generation of charge carriers and their transit through the insulation material [1]. Under high DC voltages, the space charge accumulation can be linked to the propagation of electrical trees due to local electric field enhancement. Over time, this may finally cause an electric breakdown [2,3]. Therefore, suppressing space charge accumulation has become a key issue in HVDC cable insulation development [4]. 

Considering the concept of environmental protection and sustainable development, polypropylene as a recyclable polymer has shown great potential for new polymeric insulation materials. Due to the poor flexibility of polypropylene its blends with polyolefin elastomers (such as poly(ethylene-co-octene), POE, has drawn a lot of attention, especially for developing the next-generation thermoplastic insulation in HVDC cables. Although PP/POE blends exhibit improved flexibility in comparison to PP [5,6] they are still facing the same problems of space charge accumulation [6]. The higher level of space charge accumulation contributes to the deformation of the electrical field. Based on one study the dependence is linear [7]. Therefore, it is important to suppress space charge accumulation.

There are many different methods to suppress space charge accumulation. It is well known that the cleaning process before compound mixing is crucial for lowering the amount of contaminations and residues which can cause increased internal charges. Moreover, designing a semi-conductive layer between the conductor and the main insulation layer allows to control the charge injection into the bulk of the insulating layer [1]. It is also reported that modification of the insulating polymer by tailoring its molecular structure can also suppress the space charge accumulation. For example, grafting maleic anhydride (MAH) with its polar carbonyl groups onto PP resulted in effective suppression of the space charge injection and accumulation [8]. Another approach is to incorporate nanoparticles into the polymer matrix [9,10,11]. Improved insulating properties (e.g., dielectric strength and voltage endurance) of these nanoparticle-filled polymers were reported [10,12]. The high- performance dielectric properties of polymer nanocomposites (PNCs) are explained by a large and evenly distributed polymer/filler interface area in well-dispersed nanoparticle-polymer systems. The large interfacial area may bring about changes in charge mobility and trapping properties, hence modifying the space charge accumulation of the bulk of a PNC insulation [13,14]. 

However, the nanoparticles tend to cluster together due to their high polarity and tendency to form hydrogen bonds. A low degree of dispersion will decrease the performance of nano-dielectric composites. To overcome this problem, surface modification of the nanoparticles is crucial in order to increase the compatibility of the filler with the polymer matrix and to prevent filler aggregation and agglomeration [15]. The most widely used method of surface modification of nanoparticles is a suspension technique: The reaction takes place between the low molecular weight modifier in the solution and the active moieties on the surface of the suspended nanoparticles in various temperature ranges. However, this method is environmentally questionable as it is solvent-based. Alternatively, a plasma technique can be utilized for surface modification of nanoparticles. This is an environmentally friendly, physicochemical process of changing the surface characteristics by a one-step technology and it virtually generates no waste [16]. Plasma modification has attracted considerable interest as a very “smart” and flexible way to treat materials which changes only the surface properties of the substrate without compromising its bulk properties [17,18]. Generally, plasma treatment creates a very thin and uniform deposit, and thus allows chemical modification of many different surfaces such as metals [19], polymers [18], textiles [20], composites, ceramics [21], or powder nano-fillers [22].

In this study, a commercial, low moisture content fumed silica was selected to be incorporated into PP/POE blends in order to enhance their electric properties. A custom-designed vertical plasma reactor was used to perform the low-pressure plasma modification of this silica. Acetylene was used as monomer to deposit a hydrocarbon layer on the silica surface in order to improve its compatibility with the PP/POE blends and therefore the dispersion of the silica in the polymer matrix. The goal of this modification is to suppress space charge accumulation of this composite via changing the charge trapping properties of silica/ PP/POE blends nanocomposites. 

## 2. Materials and Methods 

### 2.1. Materials

Commercial low-moisture-content fumed silica was supplied by Evonik Industries AG, (Essen, Germany). Acetylene was purchased from Linde Gas Benelux BV (Schiedam, The Netherlands). A blend of two different types of polyolefin-based polymers, polypropylene (PP), and poly(ethylene-co-octene) (POE), was used as the polymer matrix. Common antioxidants were added to the compounds to protect them from thermo-oxidative degradation during melt-processing.

### 2.2. Plasma Modification of Silica

The custom-designed radio frequency (13.56 MHz) vertical plasma reactor is shown in Figure 1. A schematic diagram is given in the left part of Figure 1a, and the actual reactor is shown in Figure 1b. This plasma reactor setup consists of four parts: Impedance matching unit;radio frequency (RF) generator;magnetic stirrer;vertical glass reactor.

2 g of the fumed silica were placed on the bottom of the vertical reactor, together with the magnetic stirrer bar to ensure homogenous modification. A vacuum of 0.33–0.34 mbar was applied, necessary for the plasma to ignite, and the acetylene monomer was introduced during a reaction time of 1 h to enable the plasma polymerization. By analyzing the results of thermogravimetric tests (TGA), the weight loss of the plasma modified samples is presented as a function of: Plasma power (100–300 Watt (W));gas flow rate (3–18 cm^3^/min).

The conditions for the acetylene plasma polymerization and deposition were optimized. After shutdown of the plasma polymerization acetylene was introduced into the reactor for another 5 min to avoid oxidation while taking out the modified sample. 

To evaluate the Acetylene Plasma Polymerization Silica Coating, thermogravimetric analysis (TGA) was performed using a Perkin-Elmer TGA-7 thermogravimetric analyzer (Waltham, MA, United States). This analysis was performed for the reference silica sample, as well as the plasma-modified silica samples. This characterization was done in a synthetic air atmosphere with a heating rate of 20 °C/min and a temperature range from ambient temperature to 850 °C.

X-ray photoelectron spectroscopy (XPS) was conducted by means of a PHI Quantera scanning X-ray microscopy and X-ray photoelectron spectroscopy from Physical Electronics GmbH (in Munchen, Germany), It is based on irradiating a material with a beam of X-rays, while simultaneously measuring the kinetic energy and number of electrons that escape from the surface (up to 10 nm in depth) of the material being analyzed. In this way it is possible to measure the elemental composition in a parts per thousand (ppt) range.

The scanning electron microscopy (SEM) and energy-dispersive X-ray spectroscopy (EDX) were done by means of Zeiss MERLIN HR-SEM microscope (Oberkochen, Germany). 

### 2.3. PP/POE/Silica Composite Preparation

The different nanocomposite samples were prepared by melt-blending of the reference (unmodified) silica or plasma-modified silicas with the PP/POE polymer blend (PP/POE ratio of 55:45). The material formulations are shown in Table 1. Small batches of 12 g were compounded using a Haake MiniLab Rheomex CTW5 mini twin conical screw extruder (Thermo Fisher Scientific, Waltham, MA, USA) using a compounding temperature of 230 °C, a screw speed of 100 rpm and a mixing time of 4 min. After melt-blending the compounds were immediately transferred to a Haake MiniJet Pro Piston Injection Moulding System (Thermo Fisher Scientific, Waltham, MA, USA) and injection-molded into thin sheets (size of 26 × 26 × 0.5 mm). The injection mould temperature was 60 °C, injection temperature was 230 °C, mean injection and holding pressure was 960 bar, and the total injection time (injection and holding) was 40 s.

#### 2.3.1. Characterization of PP/POE/Silica Composites

Differential scanning calorimetry (DSC) was performed using a DSC Q2000 from TA Instruments (TA Instruments, New Castle, DE, USA). Disc-shaped samples weighing 12–14 mg were cut from the injection-molded thin sheets, placed in aluminum pans, and inserted into the DSC cell. The samples were first heated from ambient temperature to 230 °C at a rate of 10 °C/min and maintained at this temperature for 5 min to erase any previous thermal history. The samples were then cooled down to −20 °C (40 °C/min.) and heated again to 230 °C at a rate of 10 °C/min.

X-ray diffraction (XRD) spectra were collected from the injection molded samples by means of a Philips X’Pert 1 X-ray diffractometer (Almelo, The Netherlands). The samples were scanned with 2θ values varying from 8 to 37° with a scanning rate of 0.05 °/8 s. Three samples were measured from each material.

SEM–EDX of the PO based composites was conducted by the Zeiss Merlin HR-SEM microscope (Oberkochen, Germany). The sample was first cut into a smaller piece and embedded in epoxy resin. After solidification of the resin, the sample was polished to expose the cross-sectional surface of the embedded sample, followed by sputter-coating of the polished surface with Au. The scanning electron microscopy (SEM) was done by Zeiss Merlin HR-SEM. The sample was prepared in liquid nitrogen without coating.

#### 2.3.2. Thermally Stimulated Depolarization Current (TSDC)

The charge trapping properties of the composites were studied by the thermally stimulated depolarization current (TSDC) technique. Circular Au electrodes (diameter 16 mm, thickness 100 nm) were deposited on both sides of the sample sheets by electron-beam evaporation under high vacuum (< 1 × 10^−6^ mbar). The TSDC measurement system consisted of a liquid N_2_-based temperature control system (Novocool by Novocontrol Technologies, Montabaur, Germany), a high voltage DC power source (Keithley 2290E-5), and a sensitive electrometer (Keithley 6517B by Keithley Instruments, Cleveland, OH, USA). The measurements were performed using a shielded sample cell equipped with a PT100 temperature sensor (Novocontrol BDS1200HV by Novocontrol Technologies, Montabaur, Germany). The current measurement sensitivity of the shielded measurement system was better than 1 pA. During high voltage application, a series resistor (100 kΩ) and a diode-based overload protection circuit were utilized to protect the electrometer in case of sample breakdown. The TSDC measurement procedure consisted of the following steps: The sample was heated from room temperature to 70 °C and stabilized for 5 min.A DC poling field of 3 kV/mm was applied for 20 min under isothermal conditions at 70 °C.The sample was rapidly cooled down to −50 °C with the voltage still applied, and kept at this temperature for 5 min for stabilization.The poling field was removed and the sample was short-circuited. The short-circuited sample was maintained at −50 °C for 3 min to allow fast polarization to decay.The sample was linearly heated up to 130 °C with a heating rate of 3 °C/min while measuring the thermally stimulated depolarization current.

In the above procedure, the electrometer was used to measure the current through the sample also during the isothermal polarization and cooling phases before the measurement of TSDC. Although the polarization time was too short to reach a steady-state DC conduction current [23], the isothermal charging current data nevertheless provided an indication of the transient electrical conductivity behavior of the sample. Further details on the isothermal polarization/thermally stimulated depolarization current measurement are presented in Appendix A.

#### 2.3.3. Dielectric Spectroscopy

Complex permittivity ε_r_^*^ of the injection molded composite samples was measured in the frequency range of 0.01 Hz to 1 MHz using Novocontrol Alpha-A dielectric analyzer (Novocontrol Technologies, Montabaur, Germany) with a ZG4 test interface (two-wire mode). For the sample capacitance range considered in this study, the absolute loss factor (tan δ) measurement accuracy was approximately ± 10^−4^ (absolute phase accuracy of 6 m°). The measurements were performed at room temperature using an AC measurement voltage of 1 Vrms. The samples were the same as those used for TSDC; the permittivity measurements were performed right before and after the TSDC measurement from the same samples.

## 3. Results and Discussion

### 3.1. Characterization of the Plasma Modified Silica

#### 3.1.1. TGA

##### Optimization of the Plasma RF Power via TGA Weight Loss Measurements

The silica samples were treated at the same gas flow rate (6 cm^3^/min) with different RF power settings for a duration of 1 h. The TGA results indicated a two-step thermal degradation kinetics of the deposited organic coating. The first step starts around 300 °C and the second around 450 °C. Further mass loss (above 600 °C) can be explained by the remaining silanol group condensation similar to the reference silica sample. The two-step decomposition kinetics indicate the presence of various hydrocarbon species on the silica surface [24,25]: Linear and branched polyacetylene chains, that degrade at lower temperatures, and highly cross-linked or possibly even carbonized structures [26] decomposing at higher temperatures. A schematic diagram of the plasma modification mechanism is shown in Figure 2. The cross-linked high molecular weight network is caused by small fragments, radicals, and atom formed in the plasma. These species recombine randomly together to form new irregular structures on silica surface.

Figure 3 shows the TGA curves of the reference sample and the plasma surface-modified samples: The weight loss percentage varies with the plasma RF power applied. Figure 4 shows a plot of the weight loss of the plasma modified silica samples in correlation to the RF power showing a non-linear relation between the applied RF power at 6 cm^3^/min flow rate and the maximum weight loss. The weight loss increases with increasing RF power up to a maximum at 150 W RF power; further increase of the plasma RF power results in a decrease of the weight loss. This non-linear correlation can be explained by an increasing electron concentration in the plasma with increasing RF power. As a consequence, the excitation and ionization reaction rates also increase which gives a higher polymerization efficiency. However, at higher levels of RF power the number of low-energy electrons becomes higher due to more frequent electron–electron and electron–neutral collisions. These collisions also lead to decomposition of larger fragments and formation of smaller and more stable fragments which suppress the degree of polymerization in the gas phase [27].

Based on these results, the RF power 150 W which gives the highest deposition level was used for the following steps. 

##### Optimization of the Gas Flow Rate via TGA Weight Loss Measurements 

Figure 5 shows the TGA curves of the reference sample and the samples modified with acetylene plasma at various gas flow rates. The modified samples were treated at the same optimized RF power (150 W) with different gas flow rates and for a duration of 1 h. The TGA results shown in Figure 5 depict an even more complex thermal decomposition behavior of the modified samples than presented in Figure 3. Figure 6 shows the plot fitted to the weight loss percentage of plasma modified silica samples versus the gas flow rate. It shows a non-linear relation with the applied flow rates at 150 W RF power, similar to the results presented in Figure 4 for different RF power values. The weight loss increases firstly, reaches a maximum value at around 8 cm^3^/min and then decreases. This phenomenon can be explained by the mean free path theory (1) [28]:(1)$$\lambda \approx \frac{1}{n\sigma_{c}\left(v\right)}$$
where *λ* is the mean free path (the distance traversed by an electron between two collisions), *n* is the particles (atoms, electrons, ions, neutrals) density, and *σ_c_*(*v*) is the particles (atoms, electrons, ions, neutrals) average cross section (the subscript *c* denotes collision). 

The flow rates vary the amount of gas introduced into the plasma system per minute. The higher the applied flow rate, the higher the ionized gas particle density in the plasma system. As a consequence, length of the free path in the plasma system becomes shorter (based on the above Equation (1)), likewise, when a lower flow rate is applied the mean free path is longer due to the lower particle density and the collision possibility between the partials (electrons, ions, neutrals) is very low. This leads to a lower ionization rate and a relatively low plasma polymerization rate. With increasing flow rate, the mean free path is decreasing, so the collision possibility is increasing. Therefore, the plasma polymerization rate is increasing. However, the increasingly more frequent collisions resulting from higher gas flow rates will lead to more energy dissipation of the plasma electrons or ionized particles, resulting in a decrease of gas particles ionization reaction.

#### 3.1.2. Analysis of the Polymer Deposit on the Silica Surface by XPS

Figure 7 shows the XPS spectra of the reference silica and plasma modified silica. The strong O 1s peak (533.2 eV) and the Si 2p peak (103.5 eV) are characteristic for silica. The energy range between 25 and 104 eV with one Si and several O peaks, can be considered a “fingerprint” of these materials. The reference silica also produces signals of C 1s (283.8 eV) due to the remaining traces of hydrocarbon contaminants introduced during the XPS sample preparation.

From Figure 8, one can observe that the spectrum of the plasma modified silica shows a significant increase of the C 1s signal, which indicates that organic groups are present on the plasma modified silica surface due to the polyacetylene deposition via plasma polymerization. 

XPS analysis for the elemental surface analysis was performed on seven samples. Table 2 shows the XPS results of the reference sample and the samples treated at three different RF power settings. Table 3 shows the XPS results of the reference sample and the samples treated at three different gas flow rates. The results of these measurements are corresponding with the TGA results: The samples with the highest weight loss in the TGA analysis exhibit the highest content of elemental carbon. This proves that the chemical deposition on the silica surface is originating from the acetylene plasma polymerization.

#### 3.1.3. Analysis of the Polymer Deposit on the Silica Surface by STEM–EDX

To confirm the XPS results and to study the morphology of the silica, the reference sample and selected plasma modified samples with high degrees of modification were characterized by scanning electron microscopy (SEM) and at the same time to acquire the spectra of energy-dispersive X-ray fluorescence spectroscopy (EDX). Figure 9 shows the EDX spectrum with the same elements as identified in the XPS spectrum in Figure 7. The EDX spectra confirm the absence of carbon on the reference silica and the presence of carbon on the plasma modified silica due to the presence of the acetylene plasma polymer on the surface of silica. 

Figure 10 shows the morphology and microstructure of the reference silica and the plasma modified silicas. The typical branched structure of the reference silica aggregate is obvious [29]. Plasma modified silica shows a less branched structure and smaller dimensions. This change in structure can be explained as follows:The electron or ionized particles in the plasma hit the silica and break weaker bonds in the silica aggregates during the plasma modification, which resulted in smaller dimensions of the plasma modified silica units.The hydrocarbon layer deposited on the silica surface after modification prevents re-aggregation of the plasma modified silica.

In addition, this aggregate of silica is only one example, but it is a general picture which shows that the appearance of the aggregates is different: The modified silica shows the same degree of brightness over the whole surface of the aggregate. The reference silica exhibits a very bright area in the upper part (closer to the electron source) and is rather dark in the lower part. It is again a strong suggestion that there is indeed polyacetylene grafted on the plasma modified silica surface. The reference silica does not conduct electrons over its surface, so they accumulate on the upper part where the radiation is stronger, giving bright spots, whereas the lower part remains darker. When a conductive polyacetylene layer is deposited on the silica surface, the electrons are distributed evenly over the whole aggregate surface which may indicate some degree of conductivity originating from conjugated double bonds. It should be noted that the plasma polymerization of acetylene will not result in a very regular structure as described in [30] but rather in distributions of sp^2^ and sp^3^ bonds [31]. In the extreme case nano diamonds can be formed with plasma vapour deposition of CH_4_/H_2_/N_2_ [32]. Assuming a maximum of 10% polymer content of modified silica of 30 nm diameter the thickness of the polymer layer may be around 0.5 nm.

### 3.2. Characterization of Silica Filled PP/POE Blends Nanocomposites

The plasma modified silica with the highest yield of polyacetylene surface deposition (P-silica) was incorporated into the PP/POE blends matrix in order to compare its properties with the neat PP/POE blends and the reference silica (R-silica)/ PP/POE blends nanocomposite.

#### 3.2.1. XRD Crystalline Structure Analysis 

Morphological variations arising from differences in the crystalline phase structure, crystallite size, and overall degree of crystallinity are expected to influence the dielectric properties of polymers. Thus, XRD analysis was performed to study the crystalline structure in detail. Representative XRD diffraction spectra of the injection moulded PP/POE-silica nanocomposites are shown in Figure 11 along with the unfilled PP/POE blend. Moreover, XRD spectra of the individual PP and POE components of the matrix polymer blend are also shown for reference (hot-pressed samples). The diffraction peaks at 2θ angles of 14.1, 16.8, 18.5, and 25.4° are characteristic of the thermodynamically stable α-form PP and respectively correspond to (110), (040), (130), and (060) crystallographic planes. A small (300) diffraction peak at 16.1° being attributable to β-form PP crystallites was also detected for all the samples [33]. In addition to the PP crystalline phase, orthorhombic PE crystals corresponding to (110) and (200) crystallographic planes were observed at 2θ angles of 21.4 and 23.4°, respectively, with these being attributable to the POE component of the polymer matrix [34]. 

The XRD diffraction spectra of the PP/POE composites were further analyzed by fitting and subtracting the amorphous background by cubic spline interpolation and thereafter fitting the remaining crystalline peaks using pseudo-Voigt profiles in MATLAB. The crystallinity index *Xc* was calculated by dividing the area under the crystalline curve by the total area under the original spectrum. Moreover, the apparent crystallite sizes were estimated by using Scherrer’s equation:(2)$$L=\frac{K\lambda }{\beta cos\theta }$$
where *L* is the mean crystallite size, *K* is a shape parameter (≈ 0.89), *λ* is the CuKα wavelength (= 1.5406 Å), *β* is the full width at half maximum, and *θ* is the diffraction angle. 

As shown in Table 4 the estimated crystallinity indices and crystallite sizes were, within experimental error, similar for all the PP/POE composites. Although nanosilica may bring about a mild increase in nucleation density (as will be discussed in the DSC section), the XRD data indicate no major differences in the crystalline phase structure or total crystallinity upon incorporation of neat or plasma modified nanosilica in the injection molded PP/POE composites. The relative β-form PP contents, approximated from the fitted crystalline peaks using the Turner–Jones β-form crystal index [35], were negligible (*k_β_* ~0.03) and showed no clear dependence on nanosilica; while silica nanoparticles have been reported to increase β-form PP crystallinity in some polymer composite systems elsewhere [36], this is apparently not the case for the materials studied here.

#### 3.2.2. Characterization of Phase Transitions by DSC

In order to further characterize the crystallinity of the composites as found by XRD, DSC tests were performed. The melting and crystallization process of neat PP/POE blends, reference silica filled PP/POE blends (R-silica filled PP/POE), and plasma modified silica filled PP/POE blends (P-silica filled PP/POE) were studied. The DSC results are shown in Figure 12 and Figure 13.

Based on the XRD results, it is clear that the crystalline phase consists of monoclinic α-crystals, orthorhombic crystals, and small amounts of β-crystals The characteristic peaks originated from PP segments correspond to α (110) at 14.1°, α (040) at 16.9°, α (130) at 18.5°, and α (150)/(060) at 25.4°. The orthorhombic crystal structure is formed by PE segments giving peaks (110) at 21.4° and (200) at 23.5° [34]. The weak signal from β-crystals (300) is located at 16.1°. This makes the results of the DSC measurements very clear showing two melting and crystallization peaks belonging to the orthorhombic PE phase and monoclinic α-PP phase shown in Figure 12 and Figure 13. The results strongly suggests that the mutual miscibility of the PP/POE blends is limited, and that polymer/polymer phase separation is most probably very significant. This indicates the existence of a large PP/POE interphase area which influences charge mobility, trapping, and bulk dielectric properties.

Figure 12 shows the melting curves of the three compounds and neat polymer matrices (PP and POE). It clearly shows two melting peaks. One melting peak belongs to the orthorhombic PE phase in POE polymer and the other one belongs to the monoclinic α-PP phase in PP polymer due to the consistent peak position compared to the neat polymer PP and POE matrix. All three samples show very similar melting curves with crystalline phase melting temperatures and comparable values of heat consumed during melting (Δ*H*) indicating a similar degree of crystallinity shown in Table 4. This is in line with the XRD measurement results showing similar crystallinity amount in the samples.

Figure 13 shows the cooling curves of the three compounds and the two neat polymer matrices (PP and POE). It also clearly shows the two cooling peaks belonging to POE and PP. The crystallization temperature of both crystalline phases of P-silica filled PP/POE are higher than that of R-silica filled PP/POE and neat PP/POE as shown in Table 5. This is due to nucleation effect of silica, which is dispersed in both PP/POE phases. However, the results of DSC cooling curves suggest that the silica is predominantly located in the PP phase characterized by higher crystallization onset temperature. Plasma modified silica exhibits better nucleating performance most probably due to its higher surface compatibility with PP/POE matrices than the unmodified, highly polar silica.

#### 3.2.3. Morphology Analysis by SEM and Filler Dispersion Analysis by SEM–EDX Mapping

SEM was applied to analyze the morphology of the composites shown in Figure 14 and SEM–EDX was applied to analyze the dispersion of the filler in the PP/POE blends matrix shown in Figure 15. For a clear investigation on the morphology of PP/POE matrix all the samples were prepared in the liquid nitrogen without any coating. 

Figure 14 shows the morphology of the three samples: Neat PP/POE (a), reference silica filled PP/POE (b), and plasma modified silica filled PP/POE (c). It is clearly shown that there are two separate polymer phases (one is smooth and the other one is rough) that form a layered structure. It is interesting to notice that the reference silica and plasma modified silica are most located in the smooth phase in Figure 14b,c. This is coherent with the DSC results, which exhibit that the silica has more pronounced influence on the crystallization of the PP phase. Therefore, it can be deducted that the smooth phase is the PP phase whereas the rough phase is the POE phase.

To evaluate the dispersion of the unmodified and plasma modified silicas in the PP/POE polymer matrix SEM–EDX elemental mapping was applied (Figure 15). Figure 15 shows the distributions of carbon (a1, b1, c1), oxygen (a2, b2, c2), and silicon (a3, b3, c3), which were probed using SEM–EDX to measure the intensity of the Kα line across the three composite samples (a), (b), and (c). The silicon and oxygen mapping images show the increase of the dispersion level for plasma modified silica. This is an expected result confirming the positive effect of the plasma modification on the silica/PO compatibility. The signals from silicon visible in the neat PP/POE sample (a3) come most likely from a silica contamination.

#### 3.2.4. Thermally Stimulated Depolarization Current (TSDC)

Figure 16 shows the TSDC spectra of the studied injection molded PP/POE composites. In principle, for non-polar polymers, the TSDC above the glass transition temperature is mostly attributable to space charge relaxation, with the temperature at peak maximum and the peak intensity being related to the depth and density of the charge traps, respectively. Each composite exhibited a main TSDC peak at ~75 °C (Peak I), a small side peak at ~108 °C (Peak II), and the onset of a third (incomplete) peak at > 120 °C (Peak III). Compared to the DSC data as given in Table 4, the above TSDC peaks seem to be related to the onset of the melting of the POE phase (approximately 50 °C), complete melting of this phase (approximately 110 °C), and the onset of the melting of the PP phase (approximately 125 °C), respectively. For calculating the trap depth and density distribution from the measured TSDC spectra, a numerical method allowing estimation of continuous trap density of states was applied [37], and the results are shown in the inset in 16. The model assumes slow re-trapping conditions and that only electrons were injected in the sample during polarization. Further details on the numerical method are presented in the Appendix A.

The main TSDC peak (Peak I) was located around 75 °C for all samples, corresponding to a trap depth of ~1.05 eV. The main TSDC peak temperature range was in good agreement with the onset of melting of the POE phase observed in DSC measurements, and can hence be attributed to the gradual relaxation of charge as the POE component softens and the POE crystals begin to melt at the PP/POE interphase. Slight differences in the TSDC peak intensity, and hence in the (apparent) trap density, were observed for the main peak (Peak I): For the sample filled with plasma modified silica, the TSDC peak intensity (trap density) was higher than that of the neat PP/POE composite, in contrast to the sample filled with the reference silica for which the peak intensity became lower than that of the neat PP/POE sample. Nevertheless, silica seems to have only a minor effect on the main peak characteristics which may indicate that the POE phase remains mostly unaffected by the silica. Larger differences between the TSDC spectra were however observed at higher temperatures in the 100–110 °C region (Peak II) close to the DSC melting peak observed for POE at 108 °C, with the plasma modified silica composite showing a significantly reduced trap density in this region. This indicates that silica is mostly contained in the PP phase, which is in line with the DSC results and SEM results, and that plasma modification of the silica reduced the deep trap density. Finally, each sample exhibited an initial rise portion of an incomplete TSDC peak at the highest temperatures close to 140 °C (Peak III), which is close to the DSC melting peak observed for PP at ~148 °C and may be attributed to relaxation of charges upon melting of PP crystallites.

Figure 17a presents the charging current behavior during the isothermal poling phase for each sample. While it is clear that the polarization phase was too short to reach steady-state DC conduction current in any of the samples, the transient currents nevertheless indicate differences between the samples with the plasma modified silica composite showing the lowest charging current density, and hence apparent conductivity at the end of the polarization period (1.1 × 10^−12^ S/m, 7.2 × 10^−13^ S/m, and 3.7 × 10^−13^ S/m for neat PP/POE, PP/POE-Reference silica, and PP/POE-Plasma silica, respectively). Furthermore, the integrated total charge at the end of the polarization phase (I+II) was compared to the total charge released during the TSDC (III) phase (Figure 17b). The amount of charge released during the thermally stimulated depolarization phase were found to be much lower for all samples than those injected during polarization. This behavior is quite typical, especially for relatively thick films [38], and can indicate that a large portion of the injected charge still remains trapped in the bulk of the composites. Nevertheless, the incorporation of plasma modified silica was found to increase the relative charge released during thermally stimulated depolarization, and this can be attributed to the reduced trap density in the high temperature region as seen in TSDC.

#### 3.2.5. Complex Permittivity

Real and imaginary parts of the permittivity data of the injection molded samples are presented in Figure 18 as a function of frequency at room temperature. The pristine injection molded samples, measured right before the TSDC experiments, are considered first (Figure 18, left column). For the unfilled PP/POE composite, the real part of the permittivity at 1 kHz was 2.73 which is slightly higher than that of pure PP (~2.25). A subtle increase in the real permittivity was observed upon incorporation of silica: The real permittivity at 1 kHz increased to 2.74 and 2.75 for reference silica and plasma silica, respectively. Such a small difference in these three composites is however negligible when considering the instrumentation accuracy. All the composites showed ultra-low dielectric loss at higher frequencies, with tan δ values in the 10^−4^ range which is typical for PP. However, at lower frequencies a considerable increase in the imaginary part of $\varepsilon_{r}^{*}$ was observed. When a material shows a non-negligible DC conductivity, the imaginary part of the complex permittivity contains both the polarization and conduction losses [39]:(3)$$\varepsilon_{r}^{*}=\varepsilon_{r}-j\varepsilon_{rT}^{\prime }=\varepsilon_{r}-j\left(\varepsilon_{r}^{\prime }+\frac{\sigma }{\omega \varepsilon_{0}}\right)$$
where $\varepsilon_{r}$ is the real permittivity, $\varepsilon_{rT}^{\prime }$ is the total loss factor, $\varepsilon_{r}^{\prime }$ is the polarization loss term, *σ* is the conductivity, ω is the angular frequency, and $\varepsilon_{0}$ is the vacuum permittivity. The increase in $\varepsilon_{rT}^{\prime }$ at low frequencies is likely due to the contribution of interfacial polarization and DC conductivity arising from the heterophasic morphology of PP/POE. The ref. silica and plasma silica composites exhibited slightly lower total loss factors at low frequencies in comparison to the neat PP/POE, which, in agreement with the isothermal charging current data (Figure 17a), may be attributed to slightly reduced conductivity due to the silica–PP/POE interface.

From the permittivity data measured after TSDC (Figure 18, right column), a significant reduction in real permittivity is observed for all the composites in comparison to the pristine samples. This is likely related to the morphological changes, e.g., melting and recrystallization of the POE phase and lamellar thickening of PP crystallites, that have taken place during the high temperature (isothermal) polarization and (thermally stimulated) depolarization phases. The changes observed in the DSC analysis for the crystallization behavior of the silica–PP/POE composites seem to also be reflected as larger real permittivity values in comparison to the unfilled PP/POE. At lower frequencies, both $\varepsilon_{r}$ and $\varepsilon_{rT}^{\prime }$ show a considerable increase in comparison to the pristine samples measured before TSDC along with morphological changes. This may be attributed to a relatively large amount of space charge still being deeply trapped in the specimens after TSDC. 

## 4. Conclusions

Low-pressure plasma polymerization was successfully applied for surface modification of silica. A layer of hydrocarbon compounds deposited on the silica surface after acetylene plasma modification leads to improved dispersibility of silica nanoparticles in a PP/POE matrix. 

Incorporation of the acetylene plasma modified silica into the PP/POE blend resulted in only slight changes of the polymer matrix crystallinity. However, the crystallization temperature of the blend containing acetylene plasma modified silica increased significantly indicating its higher nucleating properties. This effect is attributable to a higher degree of polymer–filler interaction caused by the presence of the hydrocarbon layer on the silica surface. The results also suggest that the silica is located mainly in the PP phase increasing its crystallization temperature to a higher extent than that of POE.

The charge trapping behavior of the PP/POE nanocomposites was studied by means of thermally stimulated depolarization current measurements under a moderate polarization field. TSDC results indicate that acetylene plasma modified silica changes the charge trapping properties and decreases the amount of injected charges into the PP/POE matrix and increases the percentage of released charges. This is caused by the acetylene plasma modification of the silica surface which assures better compatibility with the PP/POE matrix. Thus, the hydrocarbon layer on the silica surface after acetylene plasma modification changes the interface chemistry between silica and the polymer matrix and furthermore alters the nature of the charge trap sites. However, the permittivity measurement results showed that there might be still a relatively large amount of space charge still being deeply trapped in the specimens after TSDC. It might be the reason to explain why we see less charges being released and more charge being injected during TSDC measurements.

## Figures and Tables

**Figure 1 polymers-11-01957-f001:**
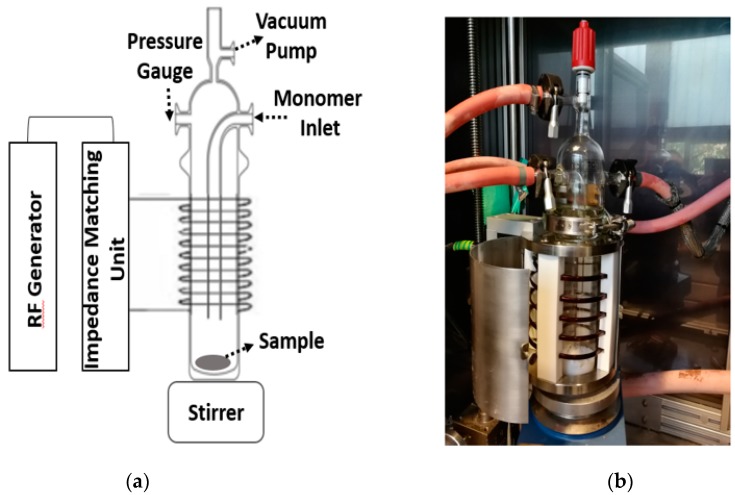
Schematic diagram of the plasma setup (**a**) and the actual reactor (**b**).

**Figure 2 polymers-11-01957-f002:**
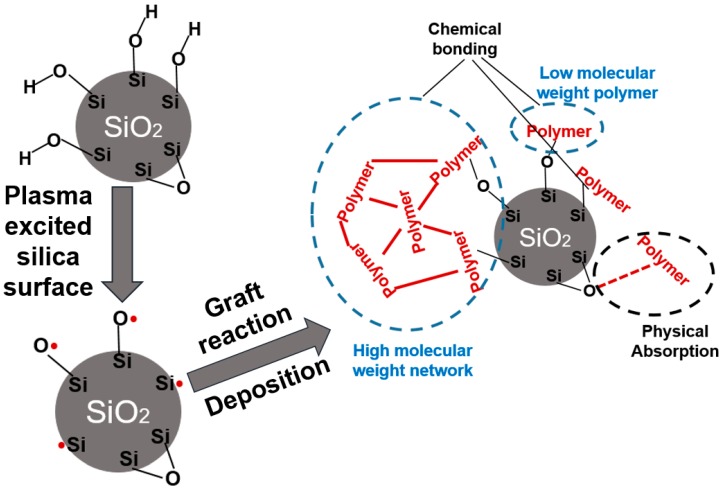
Schematic diagram of the mechanism of plasma modification of silica.

**Figure 3 polymers-11-01957-f003:**
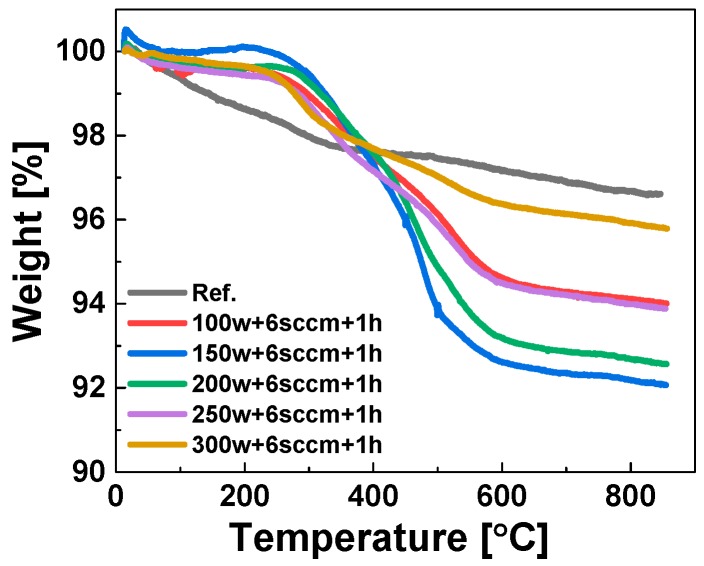
TGA results of the modified silica samples treated at different RF powers.

**Figure 4 polymers-11-01957-f004:**
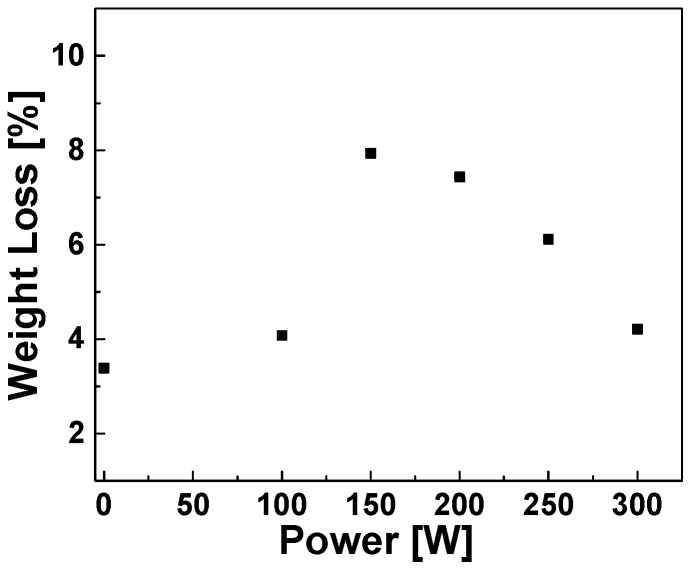
Amount of TGA weight loss correlated to radio frequency (RF) power.

**Figure 5 polymers-11-01957-f005:**
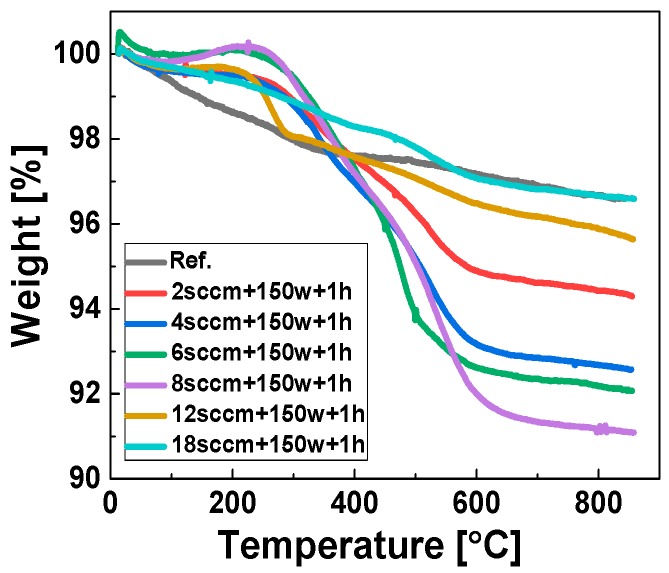
TGA results of the samples treated at different flow rates.

**Figure 6 polymers-11-01957-f006:**
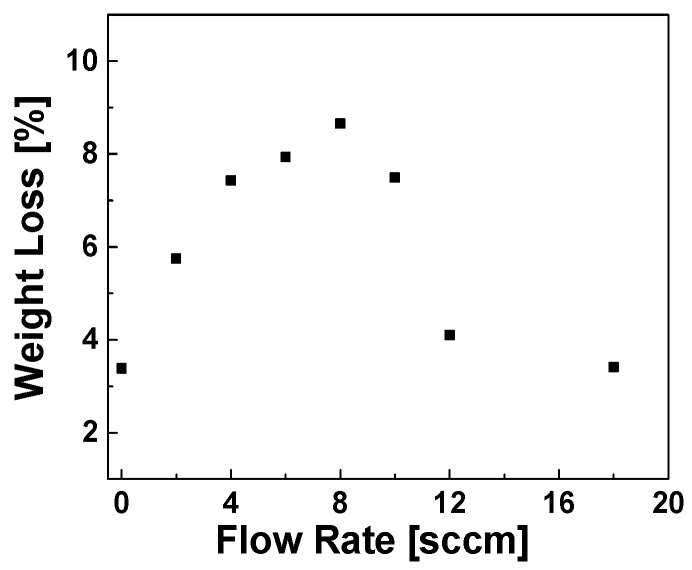
TGA weight loss compared to gas flow rates.

**Figure 7 polymers-11-01957-f007:**
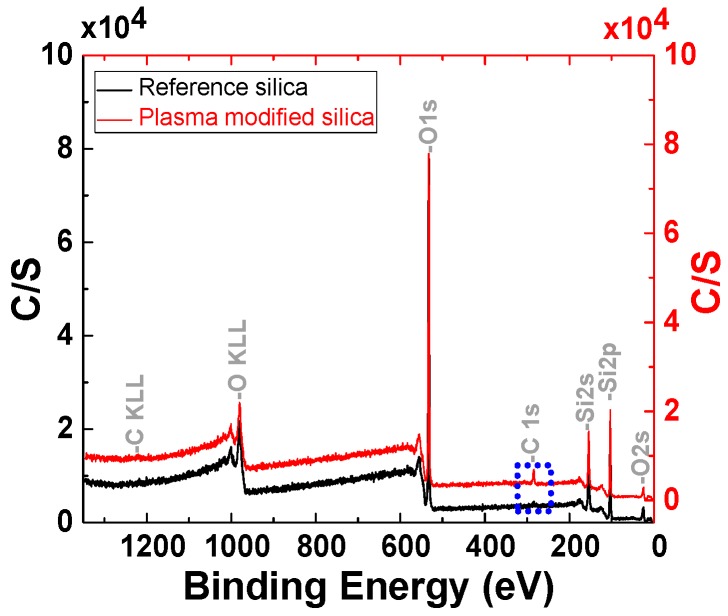
X-ray photoelectron spectroscopy (XPS) spectra of reference silica and plasma-modified silica (plasma settings: 8 sccm, 150 W, 1 h).

**Figure 8 polymers-11-01957-f008:**
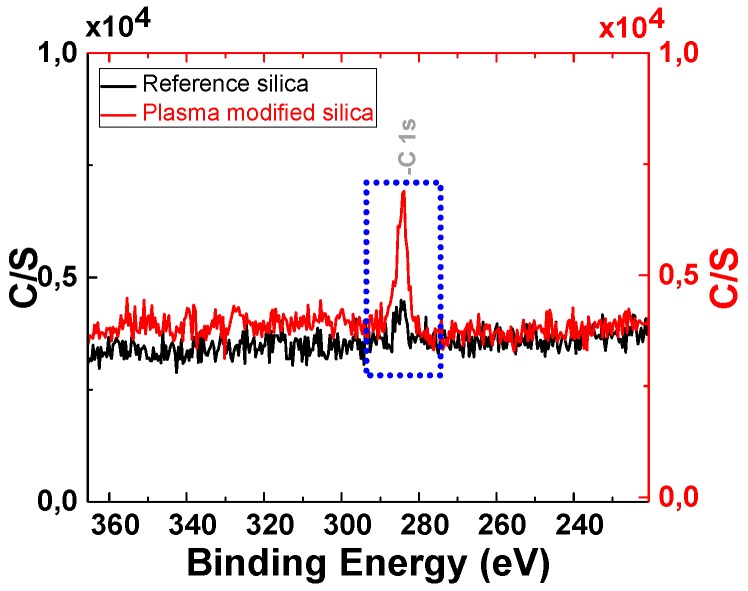
Spectra of reference silica and plasma-modified silica (plasma settings: 8 sccm, 150 W, 1 h).

**Figure 9 polymers-11-01957-f009:**
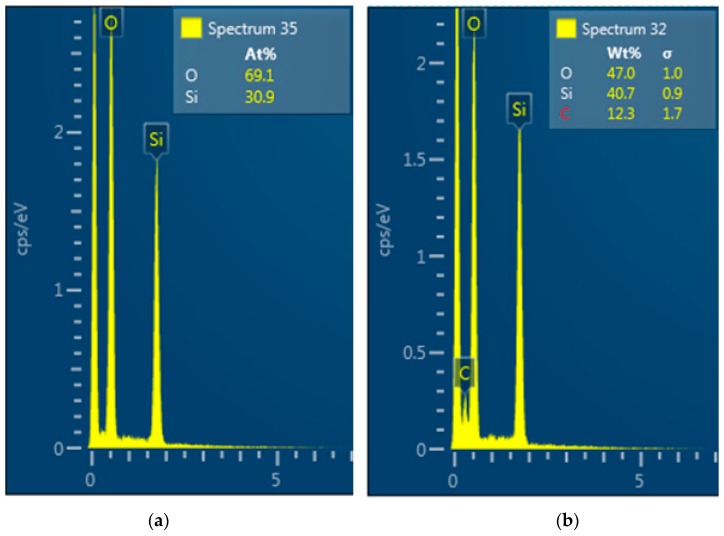
EDX spectrum of the reference silica (**a**) and plasma modified silica (**b**).

**Figure 10 polymers-11-01957-f010:**
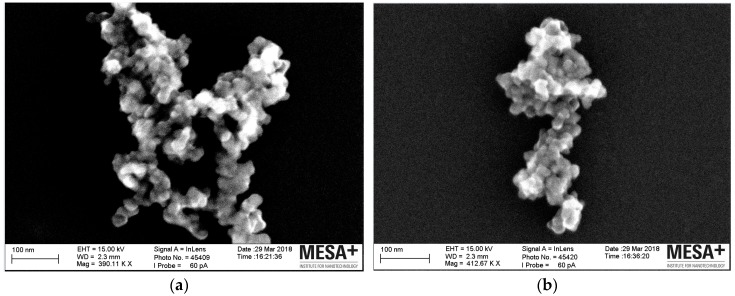
SEM image of the reference silica (**a**) and plasma modified silica (**b**).

**Figure 11 polymers-11-01957-f011:**
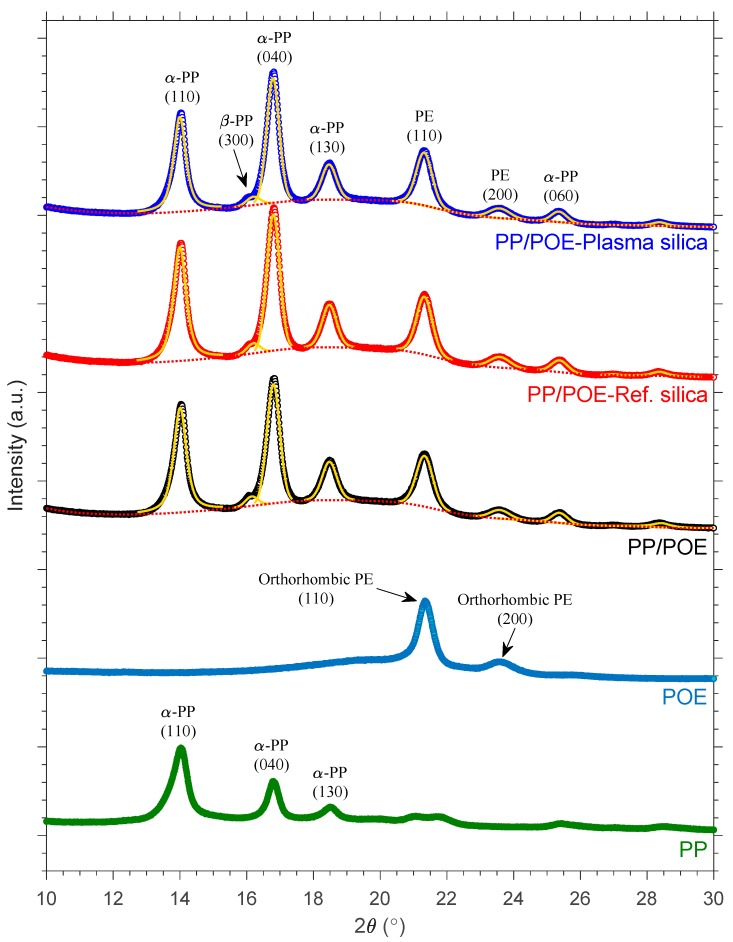
X-ray diffraction patterns of neat polypropylene/poly(ethylene-co-octene) (PP/POE) blends, reference silica filled PP/POE blends and plasma modified silica filled PP/POE blends. XRD diffraction patterns of PP and POE components are also shown (hot-pressed samples). The solid yellow lines and dashed red lines show the fitted crystalline peaks (pseudo-Voigt) and amorphous background (cubic spline) from the multi-peak fitting procedure, respectively. The curves are shifted vertically for clarity and the predominant diffraction peaks are labeled.

**Figure 12 polymers-11-01957-f012:**
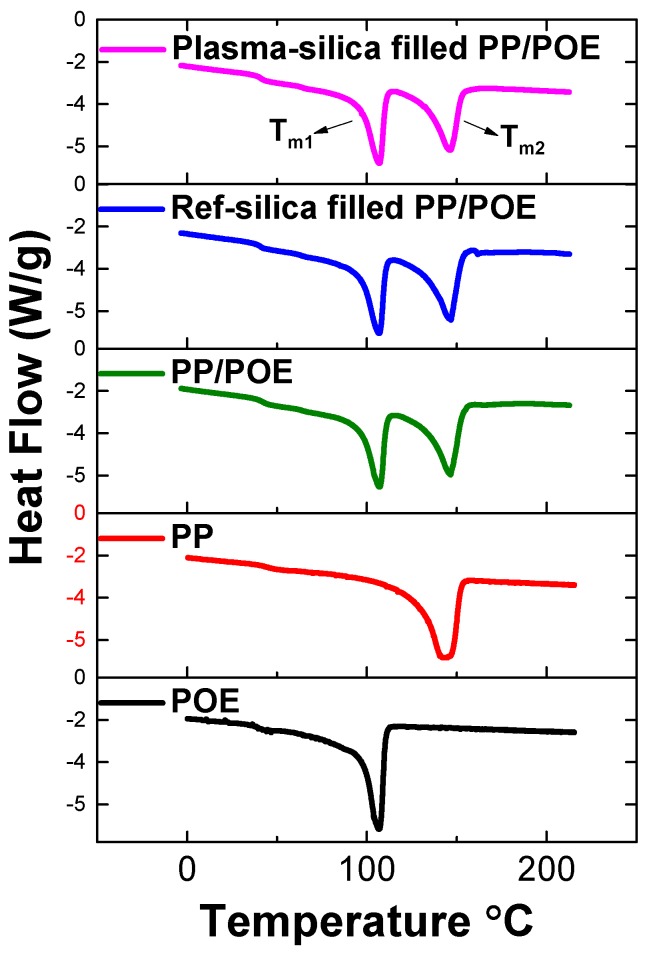
DSC melting curves.

**Figure 13 polymers-11-01957-f013:**
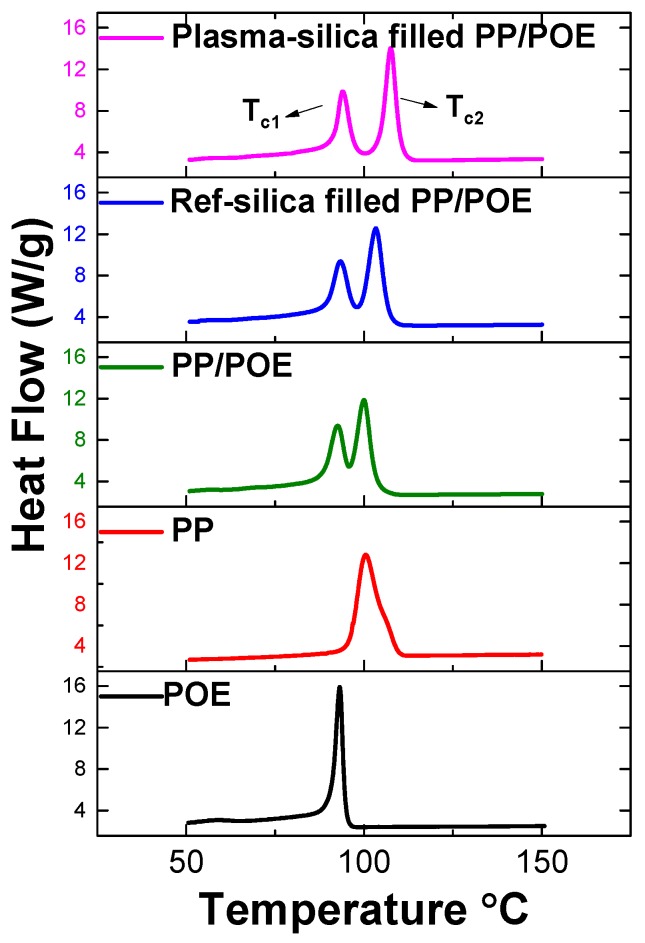
DSC cooling curves.

**Figure 14 polymers-11-01957-f014:**
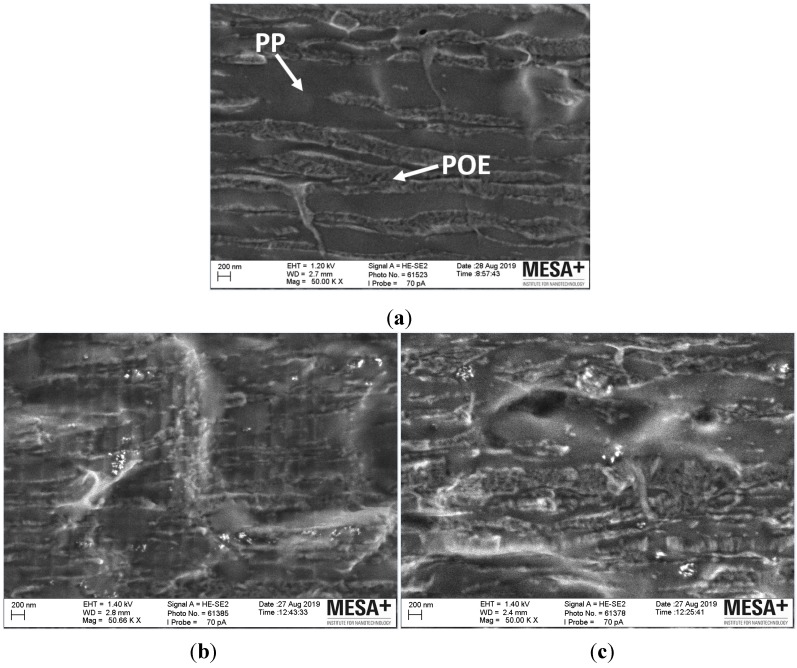
SEM images of (**a**) neat PP/POE, (**b**) reference silica filled PP/POE, and (**c**) plasma modified silica filled PP/POE.

**Figure 15 polymers-11-01957-f015:**
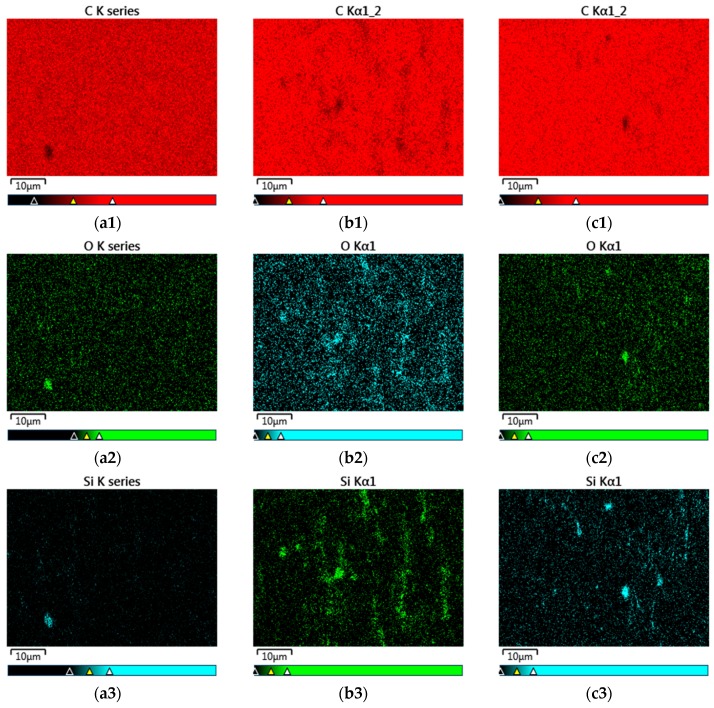
SEM–EDX Kα carbon (1), oxygen (2), and silicon (3) maps of the neat PP/POE (**a**), reference silica filled PP/POE (**b**) and plasma modified silica filled PP/POE (**c**).

**Figure 16 polymers-11-01957-f016:**
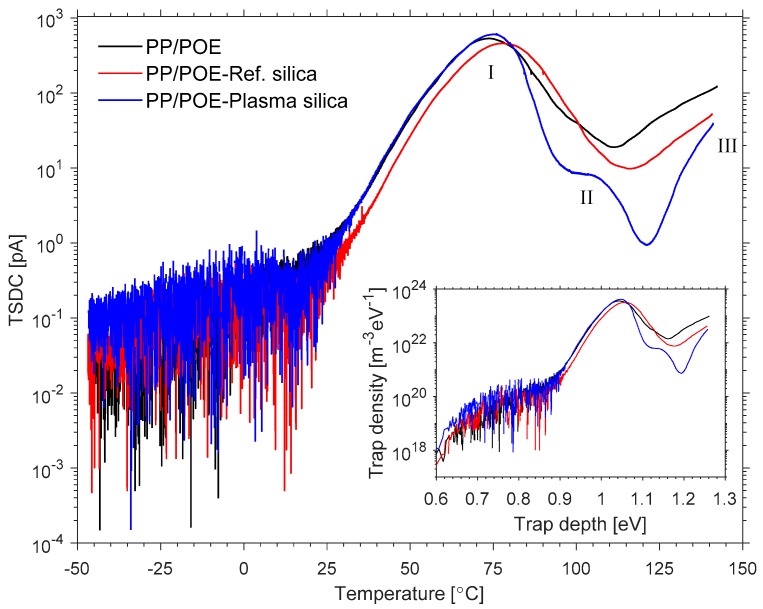
TSDC spectra of neat PP/POE, reference silica filled PP/POE, and plasma modified silica filled PP/POE. The inset shows the calculated trap depth versus density distributions.

**Figure 17 polymers-11-01957-f017:**
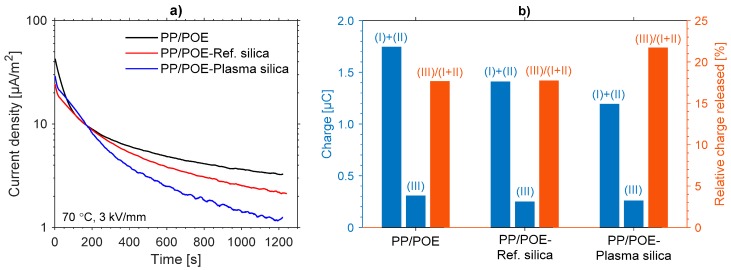
(**a**) Isothermal charging current density during the polarization phase. (**b**) Amount of charge injected (I+II) into or released (III) from the samples during TSDC measurements and their ratio (III)/(I+II).

**Figure 18 polymers-11-01957-f018:**
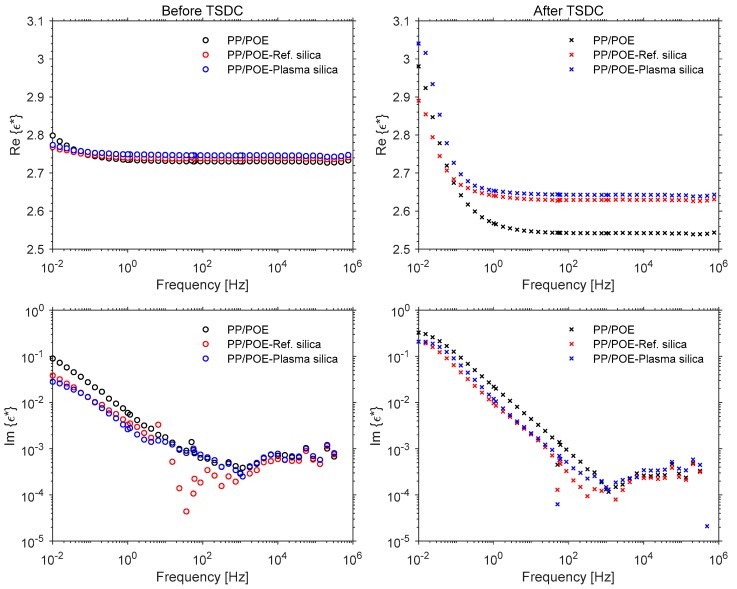
Real (**top**) and imaginary (**bottom**) parts of complex permittivity as a function of frequency before TSDC (**left column**) and after TSDC (**right column**).

**Table 1 polymers-11-01957-t001:** Material formulations.

	PP/POE	PP/POE/Reference Silica	PP/POE/Plasma Silica
PP/POE blend (55:45)	99.7%	98.7%	98.7%
Antioxidants	0.3%	0.3%	0.3%
Reference Silica	-	1%	-
Plasma modified silica	-	-	1%

**Table 2 polymers-11-01957-t002:** XPS results of the reference sample and the samples with different RF power settings.

	Element	C [%]	O [%]	Si [%]
Sample				
Reference		2.29	67.27	30.44
150 W		**4.26**	66.54	29.20
200 W		2.97	67.47	29.59
300 W		2.12	68.29	29.59

**Table 3 polymers-11-01957-t003:** XPS results of the reference sample and the samples with different flow rate settings.

	Element	C [%]	O [%]	Si [%]
Sample				
Reference		2.29	67.27	30.44
4 cm^3^/min		4.44	66.57	29.05
8 cm^3^/min		**7.92**	64.13	27.95
18 cm^3^/min		2.70	67.45	29.84

**Table 4 polymers-11-01957-t004:** Estimated XRD crystallinity index and crystallite sizes.

Material	*X_c_* [%]	Apparent Crystallite Size (nm)						
		α-PP	α-PP	α-PP	α-PP	β-PP	PE	PE
		(110)	(040)	(130)	(060)	(300)	(110)	(200)
**Neat PP/POE**	31.3 ± 2.1	17.9	17.6	15.2	14.6	16.4	13.7	11.7
PP/POE-Ref silica	32.0 ± 2.1	18.0	17.4	15.4	14.3	18.1	13.9	11.5
PP/POE-Plasma silica	31.4 ± 1.2	18.5	17.8	15.6	14.6	17.0	14.0	11.4

**Table 5 polymers-11-01957-t005:** Calculated melting and cooling parameters.

Material	Melting				Crystallization			
	*T_m1_* (°C)	Δ*H_m1_*	*T_m2_*	Δ*H_m2_*	*T_c1_*	Δ*H_c1_*	*T_c2_*	Δ*H_c2_*
Neat PP	-	-	142.5	83.0	-	-	100.5	84.7
Neat POE	107.0	64.0	-	-	93.1	56.0	-	-
Neat PP/POE	108.5	11.5	145.4	18.7	90.0	19.0	100.3	26.0
Reference silica filled PP/POE	108.5	10.9	145.6	18.3	92.7	22.3	103.8	31.4
Plasma silica filled PP/POE	107.7	11.6	144.6	18.1	93.4	25.2	107.0	34.0

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
