# Peer review of "Surface Modification of Fumed Silica by Plasma Polymerization of Acetylene for PP/POE Blends Dielectric Nanocomposites"

_polymers, 2019, doi:10.3390/polym11121957_

Round 1
Reviewer 1 Report
The paper describes a protocol for surface modification of silica to improve the dispersibility in PP/POE and the crystallization of the polymer. Furthermore, the electric insulating properties are improved by blending the modified silica particles in the polymer. Although, plasma modification of silica is well established, blends with PP/POE are not studied very much. This process shows a small improvement in electrical insulation properties, compared to pure polymer or composites with unmodified silica.
While the manuscript is well structured and most results are presented in a logical fashion, there are minor revisions/improvements necessary for making it suitable for publication:
1)There are multiple Fig.1 and a Fig. 1.5. Please check your numbering
2)Fig. 1+8+9+1.5: The figures are very blurred, please increase resolution.
3)Line 339: "As shown in Error! Reference source not found." Missing reference.
4) Fig. 19+20: I am not familiar with these measurement techniques. How large are the error of these methods?
5) The authors claim, that the silica is mainly dispersed in the PP phase based on DSC measurements. Can these findings be supported by electron microscopy? With the current SEM/EDX pictures it is impossible to see, since the resolution is not great.
Author Response
Reviewer 1
The paper describes a protocol for surface modification of silica to improve the dispersibility in PP/POE and the crystallization of the polymer. Furthermore, the electric insulating properties are improved by blending the modified silica particles in the polymer. Although, plasma modification of silica is well established, blends with PP/POE are not studied very much. This process shows a small improvement in electrical insulation properties, compared to pure polymer or composites with unmodified silica. While the manuscript is well structured and most results are presented in a logical fashion, there are minor revisions/improvements necessary for making it suitable for publication:
Review 1: There are multiple Fig.1 and a Fig. 1.5. Please check your numbering
Answer : Dear Reviewer, I have checked and changed all the incorrect numbers. Please check.
Review 2: Fig. 1+8+9+1.5: The figures are very blurred, please increase resolution.
Answer : Dear Reviewer, I changed the resolution of these figure. Please check.
Review 3: Line 339: "As shown in Error! Reference source not found." Missing reference.
Answer : Dear Reviewer, I corrected this error. Please check.
Review 4: Fig. 19+20: I am not familiar with these measurement techniques. How large are the error of these methods?
Answer : Dear Reviewer, The figure 19 and 20 have already been updated in to Figure 17 and 18.
For Fig. 17, the data presented are based on single specimens only. It is well known that morphological and microscopic (surface/volume) differences between parallel samples may result in deviation in the measured charging and discharging currents [1]. We have, however, studied the variation between parallel samples in our case, and found the deviation insignificant with respect to the differences arising from incorporation of different silica nanoparticles. The sensitivity of the measurement system used for the (isothermal) charging current measurement and the subsequent thermally stimulated depolarization current (TSDC) measurement was better than 1 pA (see the noise floor in Fig. 16). We have improved the discussion in this section accordingly.
For Fig. 18, the measurement accuracy depends mainly on the instrumentation accuracy and resolution, as well as on the capacitance (or impedance) of the sample being measured, the measurement frequency range and the surrounding environment. The measurement accuracy of the utilized impedance analyser (Novocontrol Alpha-A) is very high; it is capable of measuring dielectric loss factor tan δ down to 10−5 range (absolute phase accuracy 2 m°) in optimum conditions. For the sample capacitance range considered in this study, the absolute loss factor (tan δ) measurement accuracy was approximately ±10−4 (absolute phase accuracy of 6 m°). Thus, the real part of the complex permittivity in Fig. 18 is very precisely measured, and for the imaginary (loss) part of the complex permittivity the measurement accuracy is approximately ±10−4, with this being adequate for assessing the low-loss PP/POE-based materials.
[1] Lau KY, Vaughan AS, Chen G, Hosier IL, Holt AF. Absorption current behaviour of polyethylene/silica nanocomposites. J. Phys. Conf. Ser., vol. 472, 2013, p. 012003. doi:10.1088/1742-6596/472/1/012003.
Review 5: The authors claim, that the silica is mainly dispersed in the PP phase based on DSC measurements. Can these findings be supported by electron microscopy? With the current SEM/EDX pictures it is impossible to see, since the resolution is not great.
Answer : Dear Reviewer, I did new SEM images of these samples. They clearly showed the two phases of PP and POE polymers. And also exhibited that the silica is mostly located in the PP phase shown in Figure 14.
Reviewer 2 Report
The authors submitted the article entitled “Surface Modification of Fumed Silica by Plasma Polymerization of Acetylene for PP/POE Blends Dielectric Nanocomposites”. I would recommend that the paper could be published elsewhere. My main comments and questions are as follows:
Overall, the writing is too lengthy. Some figures should move to Supporting and shorten the relating discussions. Also, the writings should be also improved over the article. It is too trivial and difficult to read. The title seems to be incomplete. Abstract needs to re-write. In Materials, the authors need to provide more details about the PP and POE! Several figures have poor resolution and missed. NEED TO IMPROVE! The figure orders are messy. NEED TO IMPROVE! INCONSISTENT fonts in tables. Discussion in Figs. 18 and 19 are unclear. How the authors decided the peak area since the curves have very different baseline. The authors should check the format of the references. MANY INCONSISTENT! English correction by native speaker is needed since there are so many unclear descriptions and typos.
Author Response
The authors submitted the article entitled “Surface Modification of Fumed Silica by Plasma Polymerization of Acetylene for PP/POE Blends Dielectric Nanocomposites”. I would recommend that the paper could be published elsewhere. My main comments and questions are as follows:
Review 1: Overall, the writing is too lengthy. Some figures should move to Supporting and shorten the relating discussions. Also, the writings should be also improved over the article. It is too trivial and difficult to read. The title seems to be incomplete. Abstract needs to re-write. English correction by native speaker is needed since there are so many unclear descriptions and typos.
Answer : Dear reviewer, dielectric properties of polymer nano-composites arises from many factors, namely: surface properties of the nano-filler, its dispersion and distribution in the polymer matrix, its influence on the polymer crystallinity and the size of the crystallites. Only by knowing all these properties one can discuss the dielectric properties of a nano-composite. This is the reason why this article seems to be lengthy, however all the information are important when summarizing and concluding the results.
The abstract and whole content have been improved. Please check.
Review 2 : In Materials, the authors need to provide more details about the PP and POE! Several figures have poor resolution and missed. NEED TO IMPROVE! The figure orders are messy. NEED TO IMPROVE! INCONSISTENT fonts in tables.
Answer : Dear reviewer, I added the explanation of PP/POE (polypropylene and poly(ethylene-co-octane). All the imperfect figures have been checked and corrected. The fonts in all the tables has been improved. Please check.
Review 3 : Discussion in Figs. 18 and 19 are unclear. How the authors decided the peak area since the curves have very different baseline.
Answer : Dear Reviewer, The figure 18 and 19 have already been updated in to Figure 16 and 17.
Regarding Fig. 16 and the associated discussion, the determination of total charge from the thermally stimulated depolarization current (TSDC) was based on the fact that electrical current is defined as I=-dQ/dt, where I is the (thermally stimulated) current, Q is the electrical charge and t is time. Thus, the total charge released during the time interval [t1…t2] can be calculated as the area under the TSDC curve: .For the TSDC measurement, the measurement system sensitivity was better than 1 pA (see the noise floor in Fig. 16) and the baseline is practically zero in the whole temperature range (as there is no voltage applied). We have intentionally chosen to use a logarithmic y-axis in Fig. 16 to better present the fine details also in low temperature region where the TSDC intensity is low. We have improved the discussion in this section accordingly.
For Fig. 17, the data presented are based on single specimens only. It is well known that morphological and microscopic (surface/volume) differences between parallel samples may result in deviation in the measured charging and discharging currents [1]. We have, however, studied the variation between parallel samples in our case, and found the deviation insignificant with respect to the differences arising from incorporation of different silica nanoparticles. The sensitivity of the measurement system used for the (isothermal) charging current measurement and the subsequent thermally stimulated depolarization current (TSDC) measurement was better than 1 pA (see the noise floor in Fig. 17). We have improved the discussion in this section accordingly.
[1] Lau KY, Vaughan AS, Chen G, Hosier IL, Holt AF. Absorption current behaviour of polyethylene/silica nanocomposites. J. Phys. Conf. Ser., vol. 472, 2013, p. 012003. doi:10.1088/1742-6596/472/1/012003.
Review 4 : The authors should check the format of the references. MANY INCONSISTENT!
Answer : Dear reviewer, I have corrected the format of all the references. Please check.
Reviewer 3 Report
The subject discussed in the manuscript is very interesting. Modification of silica by means of plasma polymerization of acetylene is a novel method of obtaining materials with a nanofiller well dispersed in the polymer matrix.
However a number of inaccuracies can be observed:
- In the Abstract of the paper abbreviations were used which have not been expanded and explained, such as PP/POE,
- In lines 49-50 POE is referred to as a thermoplastic elastomer,
- Although it has not been explained I assume that PEO is poly(ethylene oxide). It needs to be taken into account that POE does not belong to the polyolefins group (lines 136-138). Moreover in the section 2.3 Polyolefin/silica composite preparation – the composition of the obtained materials should be presented in a table.
- In Figure 1 the scheme, apart from grafting, suggests a crosslinking reaction. It should be explained in detail.
- The numbering of the figures mentioned in text does not correspond to the actual numbering of the figures,
- In Figure 8- EDX spectrum of the reference silica (a) and plasma modified silica (b) the same image has been used twice.
- Results presented in figure 9 (SEM image of the reference silica (a) and plasma modified silica b) do not have any significance that could contribute to the discussion.
- Line 339 an obvious mistake in text can be observed :“As shown in Error! Reference source not found.”
- In Figure 11 (“Figure 11. X-ray diffraction patterns of neat PP/POE blends, reference silica filled PP/POE blends and plasma modified silica filled PP/POE blends”) PE signals have been indicated instead of POE signals.
- Authors claim that “It clearly shows that two melting peaks belong to POE and PP.” (line 377). The statement requires clarification. I can only suppose that one of them belongs to POE while the other to PP. The same misinterpretation should be avoided in the case of cooling peaks.
The results presented in the manuscript concern only one composite based on the PP/POE blend. It seems not to be sufficient to justify general conclusions in relation to all materials consisting of PP, POE and modified silica. Moreover, as indicated above, the detailed composition of the studied material is unknown.
I suggest a major revision of the manuscript.
Author Response
Reviewer 3
The subject discussed in the manuscript is very interesting. Modification of silica by means of plasma polymerization of acetylene is a novel method of obtaining materials with a nanofiller well dispersed in the polymer matrix.
However a number of inaccuracies can be observed:
Review 1 : In the Abstract of the paper abbreviations were used which have not been expanded and explained, such as PP/POE. In lines 49-50 POE is referred to as a thermoplastic elastomer. Although it has not been explained I assume that PEO is poly(ethylene oxide). It needs to be taken into account that POE does not belong to the polyolefins group (lines 136-138).
Answer : Dear Reviewer, I added the explanation of PP/POE (polypropylene and poly(ethylene-co-octane).
Review 2 : Moreover in the section 2.3 Polyolefin/silica composite preparation – the composition of the obtained materials should be presented in a table.
Answer : Dear Reviewer, I made a table 1. all the composition of the materials I used are listed in Table 1.
Review 3 : - In Figure 1 the scheme, apart from grafting, suggests a crosslinking reaction. It should be explained in detail.
Answer : Dear Reviewer, the crosslinked high molecular weight network is caused by the small fragments,redical,atom formed in the plasma. these species recombine randomly together to form new irregular structure of deposition on silica surface. I wrote these explanation in the text. Please check.
Review 4 : - The numbering of the figures mentioned in text does not correspond to the actual numbering of the figures.
Answer : Dear Reviewer, I have checked and changed all the incorrect numbers. Please check.
Review 5 : - In Figure 8- EDX spectrum of the reference silica (a) and plasma modified silica (b) the same image has been used twice.
Answer :Dear Reviewer, I have changed the incorrected image. Please check.
Review 6 : - Results presented in figure 9 (SEM image of the reference silica (a) and plasma modified silica b) do not have any significance that could contribute to the discussion.
Answer: Dear Reviewer, I did new SEM images on these samples. It clearly showed the two phase of PP and POE polymer. And it also exhibited that the silica mostly located in one PP phase shown in Figure 14.
Review 7 : - Line 339 an obvious mistake in text can be observed :“As shown in Error! Reference source not found.”
Answer: Dear Reviewer, I corrected this error. Please check.
Review 8 : - In Figure 11 (“Figure 11. X-ray diffraction patterns of neat PP/POE blends, reference silica filled PP/POE blends and plasma modified silica filled PP/POE blends”) PE signals have been indicated instead of POE signals.
Answer: Dear Reviewer, the PE peak at (110) is correct. The poly(ethylene-co-octene) POE which I used in my paper contains separate PE phase. This PE phase is the crystalline phase. Therefore, I indicated PE peak instead POE peak.
Review 9 : - Authors claim that “It clearly shows that two melting peaks belong to POE and PP.” (line 377). The statement requires clarification. I can only suppose that one of them belongs to POE while the other to PP. The same misinterpretation should be avoided in the case of cooling peaks.
Answer: Dear Reviewer, I did DSC on the three compounds and neat polymer matrices (PP and POE). It clearly shows two melting peaks. One melting peak belong to the orthorhombic PE phase in POE polymer and the other one belongs to the monoclinic α-PP phase in PP polymer due to the consist peak position compared to the neat polymer PP and POE matrix. I have already modified this in the manuscript. Please check.
The results presented in the manuscript concern only one composite based on the PP/POE blend. It seems not to be sufficient to justify general conclusions in relation to all materials consisting of PP, POE and modified silica. Moreover, as indicated above, the detailed composition of the studied material is unknown.
I suggest a major revision of the manuscript.
Round 2
Reviewer 3 Report
In comparison with the previously submitted manuscript, the manuscript in current version has been significantly improved. Presentation of the results is clear and unambiguous. However, as I have mentioned before, results presented in the manuscript concern only ONE composite based on the PP/POE blend containing only 1% of modified silica.
I am still full of doubt if it is sufficient to justify a general conclusions in relation to all materials consisting of PP, POE and modified silica.
Author Response
In comparison with the previously submitted manuscript, the manuscript in current version has been significantly improved. Presentation of the results is clear and unambiguous. However, as I have mentioned before, results presented in the manuscript concern only ONE composite based on the PP/POE blend containing only 1% of modified silica.
I am still full of doubt if it is sufficient to justify a general conclusions in relation to all materials consisting of PP, POE and modified silica.
Answer : Dear Reviewer, regarding to your comments, we did the following modification on our conclusion.
We decided to add 1% of silica due to the following reasons:
Based on our industrial experience, because of the very low bulk density of fumed silica nano-powder, adding higher amount than 1-2% of fumed silica into PP/POE polymer matrix is not possible by means of semi industrial extruders. The feeding rate of the nano-silica is very limited in comparison to the polymers. Therefore, in our research we adjusted to this limit having in mind applicability of the composites in industrial practice. Because of the high specific surface area of fumed silica (200 m2/g), addition of 1% fumed nano-silica already results in large interfacial area, which is crucial for the dielectric performance of PP/POE/Silica composites. Such small amounts of nano-fillers are commonly investigated in the literature concerning the dielectric composites (for example [12]).
To make the conclusion more specific to the type of silica surface modification the following changes have been made: “Surface-modified silica” has been changed to “acetylene plasma modified silica” in order to make clear that the silica modified by acetylene plasma has a particular effect on PP/POE composites.
Round 3
Reviewer 3 Report
I accept the attached explanation. I think that the manuscript can be published in its current form.